# Toward a BT.2020 green emitter through a combined multiple resonance effect and multi-lock strategy

Junyuan Liu [1], Yunhui Zhu [2] ✉, Taiju Tsuboi[1], Chao Deng[2], Weiwei Lou[2], Dan Wang[1], Tiangeng Liu[1] & Qisheng Zhang [1,3] ✉

Color-saturated green-emitting molecules with high Commission Internationale de L'Eclairage (CIE) y values have great potential applications for displays and imaging. Here, we linked the outer phenyl groups in multiple-resonance (MR)-type blue-emitting B (boron)-N (nitrogen) molecules through bonding and spiro-carbon bridges, resulting in rigid green emitters with thermally activated delayed fluorescence. The MR effect and multiple inter-locking strategy greatly suppressed the high-frequency vibrations in the molecules, which emit green light with a full-width at half-maximum of 14 nm and a CIE y value of 0.77 in cyclohexane. These were the purest green molecules with quantum efficiency and color purity that were comparable with current best quantum dots. Doping these emitters into a traditional green-emitting phosphorescence organic light-emitting diode (OLED) endowed the device with a Broadcast Service Television 2020 color-gamut, 50% improved external quantum efficiency, and an extremely high luminescence of $5.1 \times 10^5$ cd/m$^2$, making it the greenest and brightest OLED ever reported.

To better reproduce the true colors in the world, the color-gamut standard in display technology has gradually advanced from standard RGB (sRGB) to the National Television Standards Committee (NTSC), and now to the Broadcast Service Television 2020 (BT.2020) standard, with the largest color-gamut triangle coverage for Commission Internationale de L'Eclairage (CIE) 1931 color space thus far. To achieve the BT.2020 standard for each of the three primary colors, the emission maximum should be in a specific wavelength region, and the full width at half-maximum (FWHM) should be sufficiently narrow. Among the different types of luminescent materials, quantum dots (QDs) possess extremely high color purity by controlling their nanoscale size[1–3]. However, the high cost and difficulty of their process control and storage hinder the large-scale applications of QDs[4]. The design and synthesis of organic fluorescent molecules with high color purity are both attractive and challenging. The use of narrow-band emitters in organic light-emitting diode (OLED) displays can reduce light loss

caused by the color filter and/or optical microcavity[5,6], thereby improving the efficiency and stability of the displays.

Generally, the emission spectra of fluorescent molecules are broad due to structural deformation in the excited state and the molecular vibrations[7–12], which are enhanced by thermal conductivity through conjugated π-bonds[13,14]. Recently, Hatakeyama et al. reported on a new kind of thermally activated delayed fluorescence (TADF) emitters based on multiple-resonance (MR)[15–17]. The separation of the highest occupied (HOMO) and lowest unoccupied (LUMO) molecular orbitals at the atomic level resulted in small singlet-triplet exchange energy[18], as well as suppressed thermal conductivity in the excited state. The spectral FWHM value of the devices containing a blue MR-TADF emitter, ν-DABNA, reached 18 nm, which was narrower than state-of-the-art micro-light-emitting diodes and QDs light-emitting diodes, and was quite close to the blue color requirements for the BT.2020 standard[18]. Enhancing the electron-donating and

[1]MOE Key Laboratory of Macromolecular Synthesis and Functionalization, Department of Polymer Science and Engineering, Zhejiang University, Hangzhou 310027, China. [2]Zhejiang Hongwu Technology Co., Ltd., Taizhou 317100, China. [3]State Key Laboratory of Clean Energy Utilization, Zhejiang University, Hangzhou 310027, China. ✉e-mail: zhuyunhui@hwoled.com; qishengzhang@zju.edu.cn

-withdrawing capabilities of the fragments in the MR-TADF molecules can result in a bathochromic shift in the emission spectra. In 2019, researchers obtained a series of green MR-TADF emitters by introducing an electron-deficient group with fluoro-substituents into sky-blue emitting DtBuCzB (Scheme 1)[19,20]. In 2020, Duan et al. obtained a green MR-TADF emitter, AZA-BN, by introducing an electron-deficient aza-aromatics group to expand the molecular conjugation[21]. The device containing AZA-BN achieved a CIE y value of 0.69, which is currently the best performing bottom-emitting green organic light-emitting diode (OLED) to date. However, the above molecular design strategies enhanced the electronic structural changes as well as solid-state solvation in the excited state, which broadened the emission band along with the emission red-shift[19-22].

In this work, we reported on two new green MR-TADF molecules, tCzphB-Fl and tCzphB-Ph (Fig. 1), with emissions that can meet the BT.2020 color-gamut standard for photoluminescence (PL) and electroluminescence (EL). These two molecules were evolved from the blue MR-TADF emitter t-DABNA and the sky-blue emitter DtBuCzB[20,23]. t-DABNA contains two isolated phenyl groups that are perpendicular to the BN-core plane. Wang et al. enlarged the π-conjugation length of t-DABNA by directly linking the phenyl groups and BN core through C-C bonds and obtained sky-blue emitting DtBuCzB[20]. However, the two outer phenyl-rings were twisted slightly out of the core plane in DtBuCzB, due to steric repulsion between the adjacent H-atoms. In this work, two more planar green-light emitting molecules were obtained by locking the outer phenyl-rings with the central phenyl-ring in DtBuCzB using spiro-carbon formation, which differed from the previous design strategy through the introduction of electron-withdrawing groups. This new strategy suppressed the excited-state distortion and the vibration modes caused by the strong intramolecular charge transfer (ICT) effect[7]. The spectral FWHM values of these two molecules in cyclohexane solution were as narrow as 14 nm. The OLEDs containing the new emitters achieved a purely green emission with an FWHM value of 24 nm and CIE coordinates of (0.21, 0.75), and this was the purest green emission reported for bottom-emitting OLEDs. Due to the TADF decay pathway for triplet excitons, the devices achieved a very high external quantum efficiency (EQE) of 31%, which was 50% higher than the control devices based on classic green phosphorescent materials.

## Results
### Photophysical properties
tCzphB-Ph and tCzphB-Fl were prepared with good yields using two-step cascade reactions from 1,5-dibromo-2,4-difluorobenzene and 9,9′-

Fig. 1 | **Structures of molecules.** Scheme showing the molecular design strategy and molecular structures of t-DABNA, DtBuCzB, tCzphB-Ph, and tCzphB-Fl.

(4,6-dibromo-1,3-phenylene)bis-9H-carbazole, as shown in Supplementary Fig. 1. These two compounds showed very small Stokes shifts of 15 nm in toluene solution ($1 \times 10^{-5}$ M) at room temperature (RT) (Fig. 2a), implying very small geometric distortion in the emissive state. The emission maxima ($\lambda$) of tCzphB-Ph and tCzphB-Fl in toluene were 524 nm and 532 nm, respectively, which considerably red-shifted compared to 483 nm for DtBuCzB (Fig. 2b). The FWHMs of the emission spectra were both 21 nm for the two tCzphBs in toluene, which was narrower than 25 nm for t-DABNA and DtBuCzB. The emission spectra of the two tCzphBs blue-shifted and became narrower with a decrease in solvent polarity (Fig. 2c and Supplementary Fig. 2). In cyclohexane (cHex), the emissions of tCzphB-Ph and tCzphB-Fl had an FWHM value of 14 nm and CIE coordinates of (0.15, 0.77) and (0.19, 0.76), respectively, and these values were comparable to the best green-light emitting QDs[24-26]. This was also the first time the CIE y value of the PL spectrum of a molecule exceeded the NTSC green-light standard of 0.71[19-21,27-31]. An analysis for the appearance of the narrow band in tCzphB-Ph and tCzphB-Fl is given using the Huang-Rhys factor (Supplementary Note 1). Another analysis on the observation of wider PL band of tCzphB-Ph and tCzphB-Fl with FWHM of 21 nm in toluene compared with FWHM of 14 nm in cHex is given in Supplementary Note 2.

By doping into a classical host material 4,4′-bis(N-carbazolyl) −1,1′-biphenyl (CBP) at a concentration of 2 wt%, tCzphB-Ph, and tCzphB-Fl exhibited their emission maxima at 529 nm and 537 nm with FWHM values of 30 and 33 nm, respectively (Fig. 2c and Supplementary Fig. 2). The FWHM values of these two compounds in the carbazole-based host were considerably broader than in toluene, which was associated with the polarity of the carbazole units and the inhomogeneity of molecular aggregation[32]. To inhibit the effect of solid-state solvation, a low polar sphere-like molecule, 2,2′,7,7′-tetraphenyl-9,9′-spirobi [thioxanthene] (TPSS), was used to replace the commonly used carbazole derivatives as the host. The emission maxima of tCzphB-Ph and tCzphB-Fl in the TPSS host were comparable to those in CBP, and the FWHM values were reduced to 23 and 25 nm, respectively (Fig. 2c). Because the highly rigid molecular frameworks suppressed structural deformation-induced nonradiative decay, the PL quantum yields (PLQYs, $\Phi$) of tCzphB-Ph and tCzphB-Fl doped into the TPSS films (2 wt%) reached 0.98 and 0.93, respectively. In addition to the fluorescence decay with a lifetime ($\tau$) of approximately 7 ns, a TADF component with a lifetime of hundreds of microseconds was observed (Fig. 2d and Supplementary Fig. 3). The TADF lifetimes are much longer than those of a few microseconds observed in traditional ICT-type TADF emitters with similar $\Delta E_{ST}$ (-0.04 eV, Table 1)[18], which can be ascribed to the small Huang-Rhys factors of these tCzphBs according to the Marcus–Levich–Jortner theory (Supplementary Note 3).

Single crystals of tCzphB-Ph and tCzphB-Fl were successfully obtained by slow methanol diffusion in dilute dichloride solutions. The crystal structures shown in Fig. 3a and Supplementary Fig. 4 revealed a planar skeleton in both molecules. The angles between the central phenyl ring and the surrounding phenyl rings were nearly 0 degrees. The phenyl and fluorenyl groups that connected to the spiro-carbons were almost perpendicular to the BN plane. Steric hindrance provided by these groups separated the BN planes of the adjacent molecules by a distance of 7.3 Å, which inhibited the π-π interaction-induced fluorescence quenching and spectral broadening. The natural transition orbital (NTO) pairs for the first excited singlet ($S_1$) state were simulated by time-dependent density functional theory (TD-DFT) at the PBE0/6-31G (d, p) level[33]. As shown in Fig. 3b and Supplementary Fig. 5, the highest occupied (HONTO) and lowest unoccupied (LUNTO) NTOs of the tCzphBs were spatially separated, the same as those of t-DABNA and DtBuCzB. However, the delocalization range of the orbitals in the tCzphBs was larger than those in t-DABNA and DtBuCzB. The vertical emission energy of tCzphB-Ph was calculated to be 2.53 eV, which was

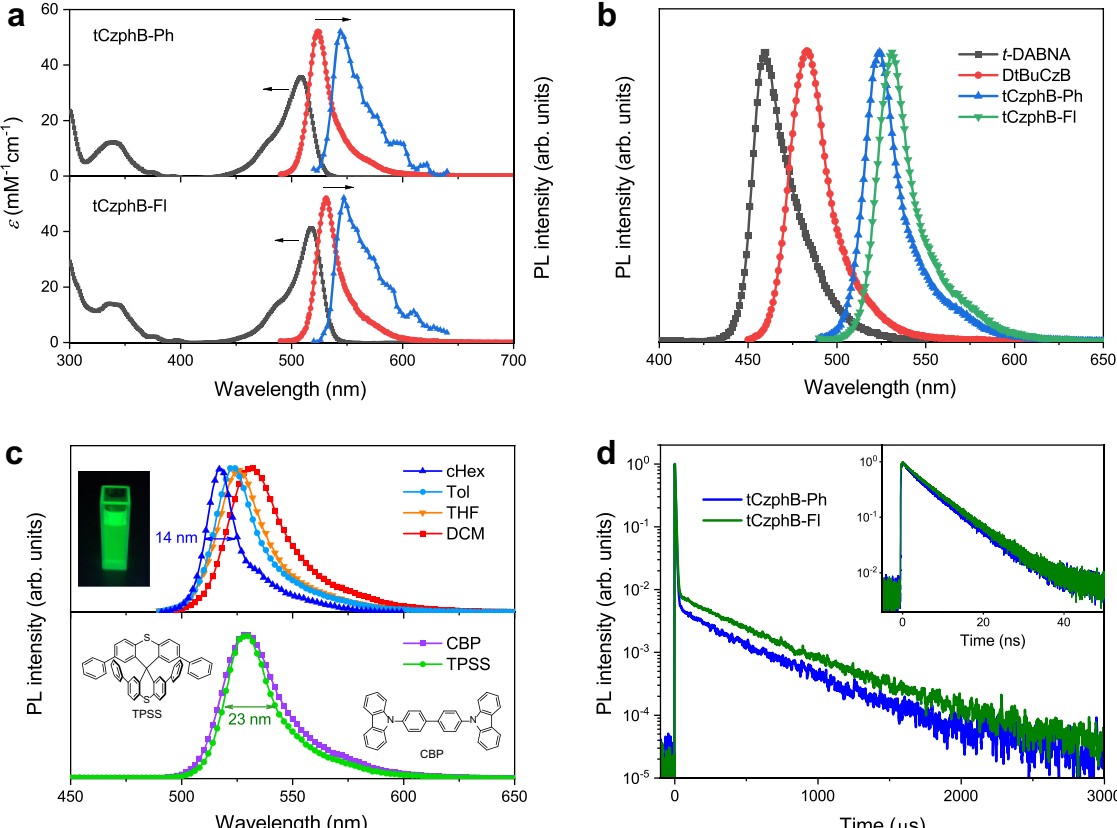

**Fig. 2 | Absorption, emission, and transient decay spectra. a** UV-vis absorption (gray line) and fluorescence (red line) spectra in $10^{-5}$ M toluene at RT and phosphorescence (100–1000 μs component, blue line) spectra in toluene at 77 K. **b** Fluorescence spectra of *t*-DABNA, DtBuCzB, tCzphB-Ph, and tCzphB-Fl in toluene at RT. **c** Fluorescence spectra of tCzphB-Ph in cyclohexane (cHex), toluene (Tol),

tetrahydrofuran (THF), dichloromethane (DCM), and organic semiconductor films of CBP and TPSS at RT (inset: image of cyclohexane solution under UV light, and molecular structures of CBP and TPSS). **d** Transient PL decay spectra of tCzphB-Ph and tCzphB-Fl doped into the TPSS films (2 wt%) at RT.

lower than the values of 2.96 eV for *t*-DABNA and 2.79 eV for DtBuCzB (Supplementary Fig. 6). Because the electron-donating capacity of the fluorenyl group in tCzphB-Fl was higher than the isolated phenyl groups in tCzphB-Ph, the theoretical transition energy of tCzphB-Fl was slightly lower than tCzphB-Ph, which was consistent with the experimental results.

It is known that the weak electron-vibration coupling is responsible to decrease the FWHM of the emission band when large-amplitude nuclear rearrangement in the $S_1$ state has been suppressed[10–13]. To better understand the excellent color purity achieved by the tCzphBs, the vibrationally-resolved fluorescence spectra were simulated by TD-DFT at the PBE0/6–31G (d, p) level and compared with DtBuCzB (Fig. 3c, d, and Supplementary Fig. 7)[34]. The MR-TADF molecules suppressed spectral broadening due to the stretching vibrations of the π-bonds by alternately embedding electron-rich and electron-poor atoms between the benzene rings. However, because of repulsions by the hydrogen atoms in the adjacent phenyl rings, the *tert*-butylcarbazole groups and the central phenyl ring in DtBuCzB were not in the same plane. The calculated vibrationally-resolved fluorescence spectra showed that the twisting and stretching vibrations between the *tert*-butylcarbazole moieties and the central phenyl ring in the excited state broadened the fluorescence spectrum of DtBuCzB. In the tCzphBs, the *tert*-butylcarbazole moiety and central phenyl ring were locked by a spiro-carbon atom, and consequently, vibrational relaxation in both the ground and excited states was significantly suppressed. The spectral simulations based on the Franck-Condon principle showed that the dominant vibration modes in the tCzphBs were the scissoring of phenyl or fluorenyl groups outside the main plane, and the rocking of the

substituted *tert*-butyl groups, which were almost in the low-frequency region (Supplementary Table 1) and had little effect on the fluorescence spectrum. The commonly used simulation method derives all vibration lines by Gaussian function with the same FWHM. However, the rigid tCzphBs exhibit a relatively steep vibrational potential energy curve compared to DtBuCzB (Supplementary Fig. 8), implying a narrower conformational distribution in the ground and excited states. In Franck-Condon analyses of emission spectra, the FWHM used to resolve the major vibrational band for the tCzphBs should be narrower than that for DtBuCzB, which must further enlarge the difference between the FWHMs of the final spectra.

## OLED performance

The thermal decomposition temperatures ($T_d$) of tCzphB-Ph and tCzphB-Fl were both over 400 °C, as recorded by thermogravimetric analysis (TGA) under a purged nitrogen atmosphere (Supplementary Fig. 9). The excellent thermal stability of these two compounds would allow for vacuum evaporation for device fabrication. The EL performance of the tCzphBs was then evaluated by the devices with a structure of ITO/HAT-CN (15 nm)/TAPC (70 nm)/TPSS:emitter (2 wt%, 30 nm)/Bepp₂ (5 nm)/Bepp₂:Liq (50 wt%, 30 nm)/Liq (1 nm)/Al, where HAT-CN, TAPC, Bepp₂, and Liq consisted of 1,4,5,8,9,11-hexaaza-triphenylene-hexacarbonitrile, 1,1-bis [(di-4-tolylamino) phenyl] cyclohexane, bis [2-(2-hydroxyphenyl)-pyridine] beryllium, and 8-hydroxyquinolinolato-lithium, respectively (Supplementary Figs. 10–12). As shown in Fig. 4, both devices exhibited pure green emissions with an application voltage greater than 3.0 V. The peaks, FWHMs, and CIE coordinate values of the EL emissions were 527 nm, 24 nm, and (0.21, 0.75) for the tCzphB-Ph-based device, and 535 nm,

**Table 1 | Photophysical parameters of tCzphB-Ph and tCzphB-Fl in the solutions and doped films**

| Compound | $\lambda_{abs}{}^a$ [nm] | $\lambda_{PL}{}^a$ [nm] | FWHM$^a$ [nm] | HOMO/LUMO$^b$ [eV] | $\lambda_{PL}{}^c$ [nm] | FWHM$^c$ [nm] | $\Delta E_{ST}{}^c$ [eV] | $\tau_p{}^c$ [ns] | $\tau_d{}^c$ [us] | $\Phi/\Phi_p{}^c$ | $k_F{}^d$ [×10$^8$ s$^{-1}$] |
|---|---|---|---|---|---|---|---|---|---|---|---|
| tCzphB-Ph | 508 | 523 | 21 | −5.15/−2.72 | 527 | 23 | 0.04 | 6.9 | 372 | 0.98/0.80 | 1.2 |
| tCzphB-Fl | 518 | 531 | 21 | −5.12/−2.73 | 535 | 25 | 0.04 | 7.3 | 412 | 0.93/0.81 | 1.1 |

$^a$Measured in toluene with a concentration of 10$^{-5}$ M. $^b$Determined from the cyclic voltammograms in dichloromethane and N,N-dimethylformamide for HOMO and LUMO energy levels, respectively (Supplementary Fig. 10). $^c$Measured in 2 wt% doped TPSS films.
$^d k_F = \Phi_p/\tau_p$, where $\Phi_p$ and $\tau_p$ are the individual PLQY and lifetime of the prompt component, respectively.

26 nm, and (0.26, 0.72) for the tCzphB-Fl-based device, respectively. These values exceeded the requirements for the NTSC green-light standard (CIE y = 0.71) and set a new record for the highest CIE y values among all reported bottom-emitting OLEDs (Supplementary Table 2)[19–21,27–31,35,36]. Because tCzphB-Ph and tCzphB-Fl had sufficiently small $\Delta E_{ST}$ values, enabling efficient up-conversion from $T_1$ to $S_1$, the external quantum efficiency (EQE) maximum values of their devices reached 29.3% and 26.2%, respectively. However, the devices underwent serious efficiency roll-off as the current density increased, due to the long triplet state lifetimes of the two MR-TADF emitters (Table 1). Using the tCzphB-Ph-based device as an example, EQE dropped dramatically to 13.6% and 2.6% at luminescence values of 500 and 10,000 cd/cm², respectively (Fig. 4d).

To suppress the efficiency roll-off in the MR-TADF OLEDs, phosphorescence sensitized TADF (PSTADF) OLEDs containing tCzphBs as the emitter were fabricated[37]. On the basis of the device structure mentioned above, the light-emitting layer was replaced with doped films consisting of BCz-o-TRZ: 5 wt% Ir(ppy)$_3$ and BCz-o-TRZ: 5 wt% Ir(ppy)$_3$: 2 wt% tCzphBs (structure I in Supplementary Fig. 11), where BCz-o-TRZ (9-(2-(4,6-diphenyl-1,3,5-triazin-2-yl)phenyl)−9′-phenyl-9H,9′H−3,3′-bicarbazole) and Ir(ppy)$_3$ (*fac*-tris(2-phenylpyridine)iridium) were the bipolar transport materials with a small $\Delta E_{ST}$ and a classic green phosphor, respectively[38,39]. The device without tCzphBs had a unique EL spectrum of Ir(ppy)$_3$ with an FWHM value of 58 nm and CIE coordinates of (0.26, 0.64) (Fig. 4). The device achieved an EQE maximum of 20.0% and an EQE of 18.0% at a luminance of 10,000 cd/m². The doping concentration of 5 wt% for Ir(ppy)$_3$ achieved the optimized result (Supplementary Fig. 13). After doping 2 wt% tCzphB-Ph into the emitting layer of this traditional phosphorescence OLED (PHOLED)[40], a sharp green emission from tCzphBs was detected instead of emission from Ir(ppy)$_3$, indicating an efficient Förster resonance energy transfer (FRET) from Ir(ppy)$_3$ to tCzphBs (Fig. 4a and Supplementary Fig. 14)[40]. In addition to the improved color purity with CIEy >0.71, the double-doped devices containing tCzphB-Ph and tCzphB-Fl exhibited a greatly improved EQE with a maximum of 31.3% and 29.7%, respectively, compared to the Ir(ppy)$_3$-single-doped device (Fig. 4d). Specifically, the efficiency roll-off caused by phosphor-based triplet-triplet annihilation was significantly reduced in the two double-doped device, due to the singlet harvest through a long-range FRET pathway (Supplementary Fig. S14) and the subsequent fast radiative decay ($k_F = 1.2 \times 10^8$ s$^{-1}$, Table 1)[21]. In addition, the small energy difference between the frontier orbitals of BCz-o-TRZ host (−5.26 eV/ −2.64 eV, ref. 38) and the tCzphBs (Table 1) avoids charge carrier trapping and direct triplet exciton formation at the terminal dopant. The tCzphB-Ph-based PSTADF OLED maintained a high EQE value of 30.6% at a luminance of 10,000 cd/m² and achieved an ultrahigh luminescence of $5.1 \times 10^5$ cd/m², which was associated with a high EQE of 25.9%. These were the best performance results for green-emitting OLEDs without the use of light out-coupling enhancement techniques (Supplementary Table 2). The increased EQE with respect to the control Ir(ppy)$_3$-based device was more closely related to the increased PLQY of the emitting layer. The PLQY values were 0.98 and 0.93 for the double-doped films containing 2 wt% tCzphB-Ph and tCzphB-Fl, respectively, versus 0.67 for the Ir(ppy)$_3$ doped BCz-o-TRZ film (5 wt%).

Some recent works on TADF-sensitized fluorescence devices indicated that doping fluorescent emitters with high fluorescence rate into the conventional TADF OLEDs can promote their operational stability by reducing the energy and lifetime of the excitons[41,42]. The reliability of the two PSTADF OLEDs was tested under a constant current density of 20 mA/cm², to compare the Ir(ppy)$_3$-based PHOLED. The times to reach 90% of the initial luminance (LT90) were 0.5, 12.0, and 8.1 h for the devices doping tCzphB-Ph, tCzphB-Fl, and only Ir(ppy)$_3$, respectively. The replacement of the TAPC layer with low glass-transition temperature ($T_g$ = 79 °C)[43] by a 60 nm N-([1,1′-biphenyl]−2-yl)-N-(9,9-dimethyl-

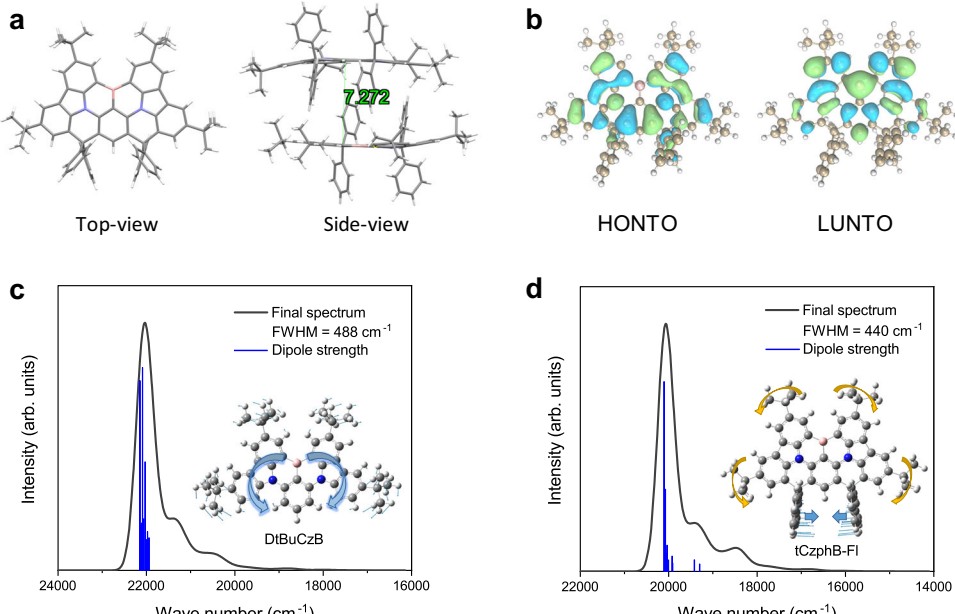

**Fig. 3 | Molecular structures and simulation results. a** Crystal structure of tCzphB-Ph. **b** HONTO (one electron) and LUNTO (one electron) distributions of tCzphB-Ph in its S$_1$ state in toluene. **c, d** Simulated vibrationally-resolved fluorescence spectra with dipole strengths of different vibrational modes for **c** DtBuCzB and **d** tCzphB-Ph. Only several modes at high energies (Supplementary Table 1) are plotted by vertical lines. The simulated fluorescence spectra envelope is derived by mean of Gaussian function with FWHM of 200 cm$^{-1}$ for each vibration line which is calculated by the TD-DFT method, to compare the spectra observed in cyclohexane. Inset: the molecular structures and domain vibration modes.

9H-fluoren-2-yl)−9,9′-spirobi[fluoren]−2-amine layer ($T_g$ = 140 °C)[44] and a 10 nm N,N-di([1,1′-biphenyl]−4-yl)−4′-(9H-carbazol-9-yl)-[1,1′-biphenyl]−4-amine layer (structure II in Supplementary Fig. 11)[45] improved the LT90 lifetimes of the above three devices to 1.4, 70.5 and 55.7 h, respectively, without slowing the device efficiency and color purity (Supplementary Fig. 16). The poor stability of tCzphB-Ph in the devices might be associated with the steric hindrance effect around the two spiro-carbon bridges which decreases the strength of the locking bonds (Supplementary Fig. 17 and Table 3). The bond dissociation energy of tCzphB-Ph was calculated to be 4.50 eV, which is markedly smaller than that of 4.63 eV for tCzphB-Fl (Supplementary Fig. 18). Although the device containing tCzphB-Fl has only an 27% longer lifetime than the Ir(ppy)$_3$-single-doped device with structure II, the initial luminance of the former (2.4 × 10$^4$ cd/m$^2$) is almost twice as higher as that of the latter (1.3 × 10$^4$ cd/m$^2$), indicating that doping tCzphB-Fl into a green PHOLED can significantly improve the device lifetime if they are measured at the same initial luminance. Fine-tuning the emission maximum of this type B-N molecule based on the structure of tCzphB-Fl is under way.

To conclude, color-saturated green-emitting molecules were successfully obtained by locking all of the phenyl rings surrounding the $sp^3$-N atoms in MR-type blue-emitting BN molecules. The combination of multiple resonance effect and rigid molecular framework suppresses both irreversible geometry relaxation and reversible stretching and rocking vibration in the excited state, which consequently minimized the Huang-Rhys factor (i.e., minimized the displacement of equilibrium position in the excited state) and narrowed the major vibrational band in the emission spectra. In non-polar solvent cyclohexane, the FWHM and CIE y values of the emission spectra were 14 nm and 0.77, respectively, and these values were comparable to state-of-the-art QDs, revealing great potential for EL and light conversion applications. The OLEDs containing these green emitters doped into a low polarity host material achieved a high EQE of 29.3%, due to the high PLQY (0.98) and TADF properties of the emitters. The device exhibited a color-saturated green emission with an FWHM value of 24 nm and CIE coordinates of (0.21, 0.75), and was the purest green bottom-emitting OLED ever reported. Doping these innovative green emitters into PHOLEDs based on classical green phosphor Ir(ppy)$_3$, can simultaneously improve the EL efficiency, color purity, and operational stability. Because phosphor-based triplet-triplet annihilation was suppressed by the FRET pathway, the PSTADF OLED exhibited a reduced efficiency roll-off compared to the control PHOLED, and achieved an extremely high luminescence over 5 × 10$^5$ cd/m$^2$. Therefore, we believe that this multi-lock strategy for MR-TADF materials could be used for the development of narrowband organic emitters with full-color emissions, as well as high EL efficiency and stability.

## Methods

### General

All commercially available reagents were used as received directly unless otherwise stated. All reactions were carried out under a nitrogen atmosphere in dry solvents. Compounds tCzphB-Ph and tCzphB-Fl were synthesized according to Supplementary Fig. 1 and purified by thermal gradient sublimation twice before device fabrication. Detailed synthesis procedures were showed in Supplementary Note 3. 2,2′,7,7′-Tetraphenyl-9,9′-spirobi [thioxanthene] (TPSS) was provided by Zhejiang Hongwu Technology Co.Ltd. The other organic materials for OLED fabrication were purchased from Jilin Optical and Electronic Materials Co. Ltd and were used as received without further purification. Nuclear magnetic resonance spectroscopy (NMR) were recorded by Bruker Avance III 400 spectrometer ($^1$H: 400 MHz, $^{13}$C: 101 MHz) and Bruker Avance III 500 spectrometer ($^1$H: 500 MHz) at 298 K. The $^1$H and $^{13}$C NMR chemical shifts were expressed as δ downfield with the internal standard of tetramethylsilane (TMS). Purity analysis was done by a Thermo Scientific Ultimate 3000 series HPLC system equipped with a RS variable wavelength detector and a Thermo Scientific column (Acclaim-120 series). Mass spectra were recorded by Thermo Scientific ISQ™ EC single quadrupole mass spectrometer using positive ion electrospray ionization. Elemental analyses were

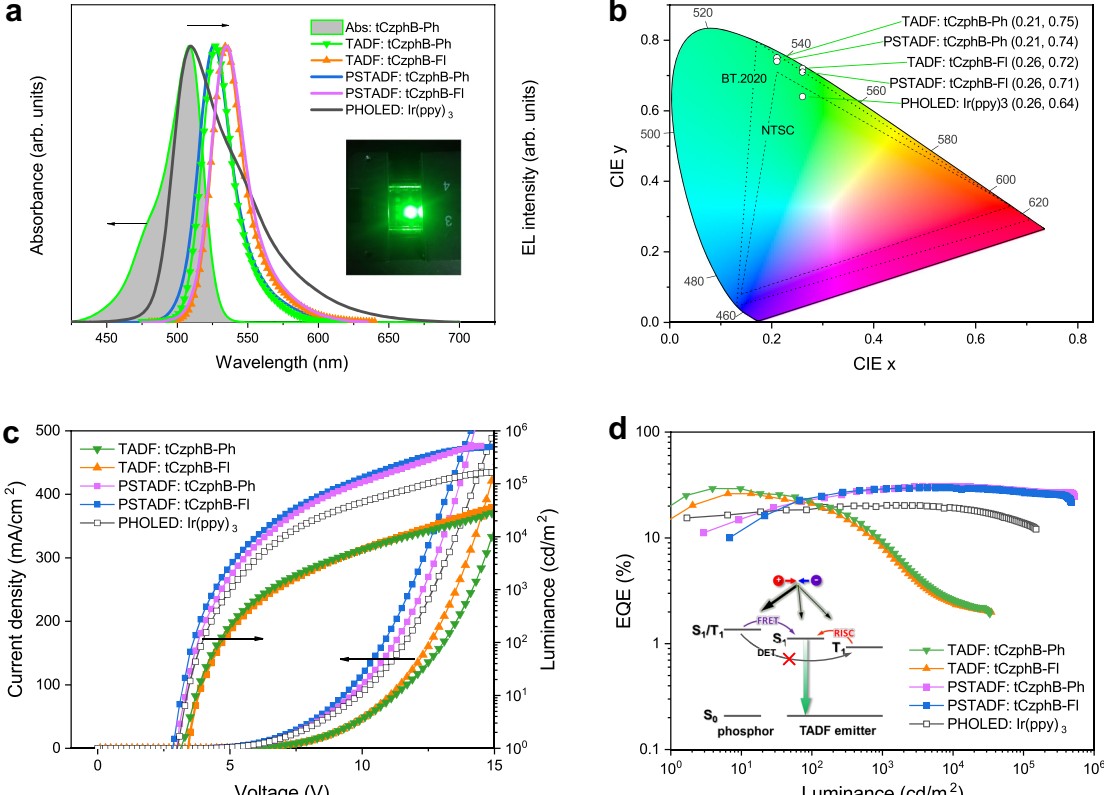

**Fig. 4 | OLED characteristics. a** EL spectra of the TADF OLEDs containing tCzphBs, PHOLED containing Ir(ppy)$_3$, and the PSTADF OLEDs containing both Ir(ppy)$_3$ and tCzphBs, to compare the absorption (Abs) spectrum of tCzphB-Ph in toluene (inset: image of the PSTADF OLED containing tCzphB-Ph). **b** Corresponding 1931 CIE coordinates of the device emission. **c** Current density-voltage-luminance characteristics of the investigated OLEDs.

**d** EQE-luminance characteristics of the investigated OLEDs. Inset: diagram of the Förster resonance energy transfer (FRET) and reverse intersystem crossing (RISC) pathways in the PSTADF OLED. The Dexter energy transfer (DET) pathway is greatly suppressed because the low concentration of the terminal dopant prevents the approach of the excited sensitizer and the terminal dopant[21].

performed using a EuroEA EA3000 and correlated with linear method.

## Quantum chemical calculations

All calculations were performed with the Gaussian 16 program package[46]. The geometries of all molecules in their ground state (S$_0$) and excited state (S$_1$) were optimized by DFT and TD-DFT, respectively, using the PBE0 function with 6-31G (d, p) basis sets in toluene. By frequency analysis, structures were confirmed at the local minima of the potential surfaces. The models of molecular structure and the HONTO/LUNTO iso-surface distribution were rendered by Multiwfn and VMD[33,47]. Based on the electronic structure in S$_0$ and S$_1$, the Laplacian bond order of each bond was calculated by Multiwfn. The bond dissociation energy for the ground state of tCzphBs was calculated according to the electron energy change in the ring-opening reaction. The single point energies of radical species were calculated at the PBE0/6-31G (d, p) level in toluene. On the basis of the optimized ground state geometry, the vibrational mode that contributes mostly to the fluorescence spectrum was selected to outline the potential energy surface. The geometric structures corresponding to the vibration mode at the amplitudes of ±1, ±0.75, ±0.5, and ±0.25 can be obtained by GaussianView software, and single-point energy calculations were performed on these structures.

## Thermal properties

Thermogravimetric analysis (TGA) were performed by TA-Q50. The TGA curve was measured at heating rate of 10 °C/min from RT to 500 °C under nitrogen flow, after eliminating residual thermal history of compound.

## Cyclic voltammetry measurements

Electrochemical properties were measured by a CHI-600E electro-chemical analyzer S-20 equipped with a glassy carbon working electrode (Φ = 5.0 mm), a platinum wire auxiliary electrode, and an Ag/Ag$^+$ reference electrode. The oxidation and reduction processes were measured by scanning the potential at a scan rate of 100 mV/s in O$_2$-free dichloromethane (DCM) and N,N-dimethylformamide (DMF), respectively, with 0.1 M tetrabutylammonium hexafluorophosphate as supporting electrolyte. The ferrocene couple Fc/Fc$^+$ was served as the external reference.

## Photophysical measurements

UV-VIS absorption spectra were measured by a Shimadzu UV2600 UV/VIS spectrophotometer. Photoluminescence spectra and quantum yield of the samples were measured using a PTI QM-40 spectro-fluorometer equipped with an integrating sphere. Time-resolved fluorescence (1 ns) and phosphorescence (0.1–1 ms) spectra and microsecond transient decay spectra were recorded by a PTI Time-Master fluorimeter equipped with a PTI nitrogen laser (GL-3300, λ = 337 nm, pulse width ≈1 ns), a ARS-4HW cryo equipment, and a closed cycle helium refrigerator. Nanosecond transient decay spectra were recorded on Horiba DeltaFlex modular lifetime measurement system patented the time-correlated single-photon counting (TCSPC) technique and equipped with a diode laser as the excitation source (λ = 370 nm, pulse width ≈50 ps).

### X-ray crystallography

X-ray diffraction data were measured at 170 K on a Bruker APEX-II CCD diffractometer using graphite-monochromated Mo-Kα radiation ($\lambda = 0.71073$ Å) from a rotating anode generator. Using the Olex2 program[48], the structures were solved with the ShelXT structure solution program using Intrinsic Phasing and refined with the ShelXL refinement package using Least Squares minimisation[49]. All Hydrogen atoms on carbons were placed in calculated positions and refined using the riding model. All non-hydrogen atoms were refined with anisotropic displacement parameters. Solvent molecules in crystal were squeezed. The crystal data as well as the details of data collection and refinement are summarized in Supplementary Tables 4 and 5.

### OLED fabrication and measurements

After carefully precleaned the indium-tin-oxide (ITO) glass substrate, the organic layers and metal layer were thermally evaporated under vacuum ($<5 \times 10^{-5}$ Pa) with 1 Å/s deposition rate for organic layers and 4 Å/s for aluminum cathode. The active area of the devices is 10 mm². The electrical characteristics of OLEDs were measured with a Keithey 2400 Source meter and a Keithey 2000 Source multimeter equipped with a calibrated silicon photodiode in a dark container. The electroluminescence spectra were recorded using a multichannel spectrometer (PMA12, Hamamatsu Photonics). The external quantum efficiency can be calculated based on the electroluminescence spectrum and electroluminescent performance when Lambertian emission was assumed. The operational lifetimes of the encapsulated OLEDs were measured by a multichannel OLED lifetime test system designed by Prof. Zhilin Zhang at Shanghai University.

## Data availability

The data supporting the findings in this study are provided in the Supplementary Information file. The single-crystal data have been deposited in the Cambridge Crystallographic Data Center database (CCDC-2175560 for tCzphB-Ph [https://doi.org/10.5517/ccdc.csd.cc2c0vcm], CCDC-2175561 for tCzphB-Fl [https://doi.org/10.5517/ccdc.csd.cc2c0vdn]). The DFT and TD-DFT raw data are available at the Figshare database (https://doi.org/10.6084/m9.figshare.20319060.v1). Source data are provided with this paper.

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

## Acknowledgements

This work was supported by the National Natural Science Foundation of China (Grant no. 51873183).

## Author contributions

Q.Z. initiated and supervised the project. J.L. and H.Z. designed, synthesized, and characterized the MR-TADF green emitters. J.L. performed the computational calculation, photophysical and electrochemical measurements of the MR-TADF emitters. J.L. and W.L. fabricated the OLEDs and measured the device performance. J.L. and Q.Z. contributed to the manuscript writing. T.T., C.D., D.W., T.L., and Q.Z. provided suggestions on experiments and writing manuscript. All authors discussed the progress of the research and reviewed the manuscript.

## Competing interests

The authors declare no competing interests.
