## [Peer Review File · Nature Communications]

REVIEWER COMMENTS

Reviewer #1 (Remarks to the Author):

The molecular structure adopted in this manuscript is an advanced version of the previous multi-resonance type materials. Though the FWHM of the green emitting MR-TADF materials was advanced, still the FWHM of the green materials is not satisfactory. The problem was resolved using the material design locking the molecular structure. The material design looks novel and this work is valuable to the OLED community.

Several concerns for this paper are as follows.

- The singlet-triplet gap is small, but the delayed fluorescence lifetime is very long. This has to be clarified. It should be short.
- How can be the EQE of the PSTADF device higher than the phosphor device itself? Detailed mechanism has to be described. When all radiative excitons of the sensitizer are transferred to the emitter, the EQE has to be the same.
- Recently, many devices show good efficiency. However, lifetime of the device is not mentioned. Lifetime is more important than efficiency. Lifetime data of the devices have to be explained.
-

Reviewer #2 (Remarks to the Author):

The manuscript claims high external quantum efficiency in green organic light-emitting diodes using fully fused emitters based on boron/nitrogen backbone. The primary selling point of the emitters is the pure green emission and the molecular design strategy is attractive. I think this work is helpful for researchers to construct highly efficient MR-TADF emitters and suitable for this journal. However, major revision is required prior to acceptance. The detailed concerns and comments are listed as follows:

1. NMR data for the emitters are not very clean. Please supply evidence of purity in the form of HPLC traces. Additionally, there are only 63 protons found in ¹H NMR for tCzphB-Ph (C₇₂H₆₅BN₂), why?

2. The Δ EST should be deduced from the prompt fluorescence and phosphorescence spectra both conducted at low temperatures. Temperature-dependent TRPL is also necessary to support the TADF nature of these two compounds. According to the lifetime and PLQY, rate constants for the emitters have to be calculated.

3. The authors suggest the partial twist between carbazole units and central phenyl ring broaden the fluorescence spectrum of DtBuCzB considerably. However, the FWHM of DtBuCzB is only 22 nm in toluene (according to the AOM paper by Wang et. al) while tCzphBs are 21 nm, the difference is therefore not that significant, please comment.

4. The authors report hyperfluorescence device using Ir(ppy)₃ as an assistant dopant. This is a detracting point to the study. It would be good if the authors were to examine devices with pure organic assistant dopants. Examples of suitable organic sensitizers are many.

5. In the phosphor sensitized devices, Dexter energy transfer (DET) may occur within the emitting layer yet this process is not shown in Figure 4d. Since the EQE roll-off is not large, DET might be suppressed, but it is necessary to further rationalize the small roll-off by more evidences. The role of the terminal dopant as fluorescence emitter or TADF emitter should be clarified.

6. Device lifetime is required to improve the quality of this work.

Reviewer #3 (Remarks to the Author):

In this study, the authors reported a novel strategy for achieving ultra-pure green emission by linking the outer phenyl groups in multiple-resonance (MR)-type B(boron)-N (nitrogen) molecules. It is impressive that the high EL efficiencies and high green color purity could be simultaneously obtained in the MR-OLEDs based on tCzphB-Ph and tCzphB-FI emitters. I think this manuscript can be accepted after minor revisions.

1. Why the two target molecules could emit ultra-pure green light ? It's better to explain the design strategy in a prominent location.

2. In Figure 3c, it seems that DtBuCzB possesses a smaller FWHM, please provide some comments on this point. In addition, to further explain the FWHM difference between these emitters, reorganization energy calculation should better be presented.
3. The orbital calculation of the excited state is necessary to show the nature of the emission.
4. The TADF characteristics such as Φ_F , Φ_{TADF} , k_{ISC} , and k_{RISC} of the novel emitters should better be calculated from the transient emissions.
5. Why the emitters show larger FWHMs in OLEDs?
6. How about of the stabilities of the OLEDs?

We gratefully acknowledge the reviewers for the time spent on our manuscript and the insightful comments that help us improve the quality of our manuscript. We answer the reviewer's comments here and highlight the changes and additions in red color in the revised manuscript.

Reviewer #1

The molecular structure adopted in this manuscript is an advanced version of the previous multi-resonance type materials. Though the FWHM of the green emitting MR-TADF materials was advanced, still the FWHM of the green materials is not satisfactory. The problem was resolved using the material design locking the molecular structure. The material design looks novel and this work is valuable to the OLED community.

Several concerns for this paper are as follows.

- The singlet-triplet gap is small, but the delayed fluorescence lifetime is very long. This has to be clarified. It should be short.

Author reply: Thank you for the suggestion. According to the Fermi's golden rule, the reversed intersystem crossing rate between the initial (T₁) and final (S₁) states is expressed by

$$k_{\text{RISC}} = (2\pi/\hbar) \langle \varphi_{\text{S}_1}(\mathbf{r}) \chi_{\text{S}_1, n}(\mathbf{Q}) | H_{\text{SO}}(\mathbf{r}) H_{\text{vib}}(\mathbf{Q}) | \varphi_{\text{T}_1}(\mathbf{r}) \chi_{\text{T}_1, n}(\mathbf{Q}) \rangle^2 \rho(E_{\text{S}_1})$$

$$k_{\text{RISC}} = (2\pi/\hbar) \langle \varphi_{\text{S}_1}(\mathbf{r}) | H_{\text{SO}}(\mathbf{r}) | \varphi_{\text{T}_1}(\mathbf{r}) \rangle^2 \rho_{\text{FCWD}}$$

where $H_{\text{SO}}(\mathbf{r})$ and $H_{\text{vib}}(\mathbf{Q})$ are the spin-orbit and electron-phonon interaction Hamiltonians, respectively, while $\varphi_{\text{S}_1}(\mathbf{r})$ and $\chi_{\text{S}_1, n}(\mathbf{Q})$ are the electronic wavefunction of the S₁ state and the vibrational wavefunction of the n -th vibrational state in S₁, respectively. $\rho(E_{\text{S}_1})$ is the density of the final S₁ state, and ρ_{FCWD} is the Frank-Condon-weighted density of states that is derived from the Franck-Condon factor $\langle \chi_{\text{S}_1, n}(\mathbf{Q}) | \chi_{\text{T}_1, n}(\mathbf{Q}) \rangle^2$. The Marcus-Levich-Jortner Theory based on the Franck-Condon Principle provides the following formula for ρ_{FCWD} :^X

$$\rho_{\text{FCWD}} = \frac{1}{\sqrt{4\pi\lambda_{\text{M}}k_{\text{B}}T}} \sum_{n=0}^{\infty} \exp(-|S_{\text{eff}}|) \frac{S_{\text{eff}}^n}{n!} \exp\left(-\frac{(\Delta E_{\text{ST}} + n\hbar\omega_{\text{eff}} + \lambda_{\text{M}})^2}{4\lambda_{\text{M}}k_{\text{B}}T}\right)$$

where S_{eff} is the effective Huang-Rhys factor of the effective mode with effective frequency ω_{eff} , which is responsible to the strength of the electron-phonon coupling, and λ_{M} is the reorganization energy, i.e., the energy required to bring the system in the S₁ to the minimum structure of the T₁ state. The theory takes into account the overlap of the vibrational state between the S₁ state with vibrational state $n=n'$ and the T₁ state with $n=n''$ where the two states have the same energy, allowing the electronic reverse intersystem crossing transition from the T₁ state to the S₁ state. The part $\exp(S_{\text{eff}})n!/S_{\text{eff}}^n$ (defined as $C(S_{\text{eff}})$) involved the Huang-Rhys factor S_{eff} gives a large value if S_{eff} is small. Both tCzphB-Ph and tCzphB-FI give $S_{\text{eff}} \approx 0.12$ (**Figure 2d and Figures S4**, respectively), and then gives the intensity for the 0-0 component ($n=0$), $C(S_{\text{eff}}=0.12) \approx 0.135$. Classical charge transfer (CT) molecules have a large S_{eff} and a large $C(S_{\text{eff}})$ as a result, e.g., $S_{\text{eff}}=2.0$ and $C(S_{\text{eff}}=2) \approx 14.8$.

Because k_{RISC} is proportional to k_{TADF} , the delayed fluorescence lifetime τ_{TADF} is

proportional to $\exp(-S_{eff})S_{eff}^n/n!$. Consequently, the two tCzphB-Ph and tCzphB-FI compounds are suggested to have much longer τ_{TADF} , about by 109.6 ($=14.8/0.135$) times longer than the classical CT-type molecules. In this way, we understand the reason why the present compounds exhibit long τ_{TADF} from the component related with the Huang-Rhys factor in the equation for ρ_{FCWD} constructed by the Marcus–Levich–Jortner theory.

The above discussion was added to the Supplementary. In the text, the reason why tCzphB-Ph and tCzphB-FI exhibit long TADF lifetime at RT was clarified as follows, “The TADF lifetimes are much longer than those of a few microseconds observed in traditional ICT-type TADF emitters with similar ΔE_{ST} (~ 0.04 eV, Table 1),¹⁸ which can be ascribed to the small Huang-Rhys factors of these tCzphBs according to the Marcus–Levich–Jortner theory (Supplementary Note 3).”

- How can be the EQE of the PSTADF device higher than the phosphor device itself? Detailed mechanism has to be described. When all radiative excitons of the sensitizer are transferred to the emitter, the EQE has to be the same.

Author reply: Because of the low radiative decay rate of phosphor, the PLQY of Ir(ppy)₃ doped into BCz-o-TRZ film (5 wt%) is only 0.67. In contrast, the PLQY of the double-doped BCz-o-TRZ films containing 2 wt% tCzphB-Ph and 5 wt% Ir(ppy)₃ can achieve nearly 1. As shown in the following figures, the lifetime evaluated from the transient decay spectrum of the double-doped BCz-o-TRZ films is close to that of the tCzphB-Ph doped into the BCz-o-TRZ film (2 wt%) but much shorter than that of the Ir(ppy)₃ doped BCz-o-TRZ film (5 wt%), indicating that the Förster resonance energy transfer from Ir(ppy)₃ to tCzphB-Ph is much faster than the radiative and non-radiative decay in Ir(ppy)₃ with a rate on the order of 10^6 s⁻¹. As a consequence, the quantum yield of the doped film is governed by the emitter rather than the host and the sensitizer in both case of PL and EL.

Supplementary Fig. 14. Steady-state (a) and transient (b) emission spectra from the BCz-o-TRZ films doped with 5 wt% Ir(ppy)₃, 2 wt% tCzphB-Ph, and both at RT. The wavelength of the excitation light was 370 nm. The PLQY of the double-doped film (0.97) is considerably higher than that of the Ir(ppy)₃-single-doped film (0.67) owing to the very efficient Förster resonance energy transfer from Ir(ppy)₃ to tCzphB-Ph.

- Recently, many devices show good efficiency. However, lifetime of the device is not mentioned. Lifetime is more important than efficiency. Lifetime data of the devices have to be explained.

Author reply: Lifetime data of the devices was added to the supplementary and discussed in the text as follows. “Some recent works on TADF sensitized fluorescence devices indicated that doping fluorescent emitters with high fluorescence rate into the conventional TADF OLEDs can promote their operational stability by reducing the energy and lifetime of the excitons.^{41,42} The reliability of the two PSTADF OLEDs was tested under a constant current density of 20 mA/cm², to compare the Ir(ppy)₃-based PHOLED. The times to reach 90% of the initial luminance (LT90) were 0.5, 12.0 and 8.1 h for the devices doping tCzphB-Ph, tCzphB-F1, and only Ir(ppy)₃, respectively. The replacement of the TAPC layer with low glass-transition temperature ($T_g = 79^\circ\text{C}$)⁴³ by a 60 nm *N*-([1,1'-biphenyl]-2-yl)-*N*-(9,9-dimethyl-9*H*-fluoren-2-yl)-9,9'-spirobi [fluoren]-2-amine layer ($T_g = 140^\circ\text{C}$)⁴⁴ and a 10 nm *N,N*-di([1,1'-biphenyl]-4-yl)-4'-(9*H*-carbazol-9-yl)-[1,1'-biphenyl]-4-amine layer (structure II in Supplementary Fig. 11)⁴⁵ improved the LT90 lifetimes of the above three devices to 1.4, 70.5 and 55.7 h, respectively, without lowering the device efficiency and color purity (Supplementary Fig. 16). The poor stability of tCzphB-Ph in the devices might be associated with the steric hindrance effect around the two spiro-carbon bridges which decreases the strength of the locking bonds (Supplementary Fig. 17 and Table 3). The bond dissociation energy of tCzphB-Ph was calculated to be 4.50 eV, which is markedly smaller than that of 4.63 eV for tCzphB-F1 (Supplementary Fig. 18). Although the device containing tCzphB-F1 has only an 27% longer lifetime than the Ir(ppy)₃-single-doped device with structure II, the initial luminance of the former (2.4×10^4 cd/m²) is almost twice as higher as that of the latter (1.3×10^4 cd/m²), indicating that doping tCzphB-F1 into a green PHOLED can significantly improve the device lifetime if they are measured at the same initial luminance. Fine tuning the emission maximum of this type B-N molecule based on the structure of tCzphB-F1 is under way.”

Reviewer #2

The manuscript claims high external quantum efficiency in green organic light-emitting diodes using fully fused emitters based on boron/nitrogen backbone. The primary selling point of the emitters is the pure green emission and the molecular design strategy is attractive. I think this work is helpful for researchers to construct highly efficient MR-TADF emitters and suitable for this journal. However, major revision is required prior to acceptance. The detailed concerns and comments are listed as follows:

1. NMR data for the emitters are not very clean. Please supply evidence of purity in the form of HPLC traces. Additionally, there are only 63 protons found in ¹H NMR for tCzphB-Ph (C₇₂H₆₅BN₂), why?

Author reply: Thank you for your reminder. The deuterated solvent of CDCl₃ was

replaced by CD₂Cl₂ in the ¹H NMR measurement of tCzphB-Ph. From the updated ¹H NMR spectrum (Supplementary Fig. 19), the protons in tCzphB-Ph were found to be 65 which are in agreement with the theoretical value. The HPLC spectra of the tCzphBs are shown in Supplementary Fig. 25 and 26. A purity of 99.2% for tCzphB-Ph and 99.3% for tCzphB-FI were identified.

2. The ΔE_{ST} should be deduced from the prompt fluorescence and phosphorescence spectra both conducted at low temperatures. Temperature-dependent TRPL is also necessary to support the TADF nature of these two compounds. According to the lifetime and PLQY, rate constants for the emitters have to be calculated.

Author reply: Thanks for the suggestion. The prompt fluorescence and phosphorescence spectra measured at 50 K and the temperature-dependent transient PL decay spectra were shown in Supplementary Fig. 3. On the basis of the individual lifetime and PLQY for the prompt and delayed components, the fluorescence rate were calculated to be $1.2 \times 10^8 \text{ s}^{-1}$ for tCzphB-Ph and $1.1 \times 10^8 \text{ s}^{-1}$ for tCzphB-FI, as listed in Table 1.

3. The authors suggest the partial twist between carbazole units and central phenyl ring broaden the fluorescence spectrum of DtBuCzB considerably. However, the FWHM of DtBuCzB is only 22 nm in toluene (according to the AOM paper by Wang et. al) while tCzphBs are 21 nm, the difference is therefore not that significant, please comment.

Author reply: There is a system error between different instruments and methods. The FWHM of the PL spectrum of DtBuCzB in toluene was found to be 25 nm in our laboratory. Additionally, DtBuCzB emits sky-blue light. Comparing the spectral FWHM of these emitters in an energy unit would enlarge the difference between tCzphBs (0.099 eV for tCzphB-Ph and 0.094 eV for tCzphB-FI) and DtBuCzB (0.137eV).

4. The authors report hyperfluorescence device using Ir(ppy)₃ as an assistant dopant. This is a detracting point to the study. It would be good if the authors were to examine devices with pure organic assistant dopants. Examples of suitable organic sensitizers are many.

Author reply: Thanks for the suggestion. Actually, we have tried to use TADF emitters to be the sensitizer at the beginning, but the results were not as good as the PSTADF OLEDs reported in this manuscript (please see following figures). Because these data are not a fully optimized result, we did not show them in this manuscript. The best performance achieved by using Ir(ppy)₃ as the sensitizer, probably because Ir(ppy)₃ has the ultrafast RISC rate from the lower substate with more triplet nature to the upper substate with more singlet nature and because the small Stokes shift of Ir(ppy)₃ allowing the utilization of a narrow-bandgap host like BCz-o-TRZ.

EL spectra (a), current density–voltage characteristics (b), luminance–voltage characteristics (c) and EQE–current density characteristics (d) of the devices with a structure of ITO/HAT-CN (15 nm)/TAPC (70 nm)/EML (30 nm)/Bepp₂ (5 nm)/Bepp₂:Liq (5 wt%, 30 nm)/Liq (1 nm)/Al, where EML is BCz-o-TRZ: 10 wt% 4CzIPN or BCz-o-TRZ: 10 wt% 4CzIPN: 2 wt% tCzphB-FI.

5. In the phosphor sensitized devices, Dexter energy transfer (DET) may occur within the emitting layer yet this process is not shown in Figure 4d. Since the EQE roll-off is not large, DET might be suppressed, but it is necessary to further rationalize the small roll-off by more evidences. The role of the terminal dopant as fluorescence emitter or TADF emitter should be clarified.

Author reply: Thanks for the suggestion. It has been demonstrated in some previous works that the DET process from the sensitizer to the terminal dopant can be suppressed because the low concentration of the terminal dopant prevents the approach of the excited sensitizer and the terminal dopant (Angew. Chem. Int. Ed. 2020, 59, 17499). We cited the related references in the text and rationalized the suppression of DET in Fig. 4d. The sentence “Specifically, the efficiency roll-off caused by phosphor-based triplet-triplet annihilation was significantly reduced in this double-doped device, due to the singlet harvest through a long-range FRET pathway (Fig. 4d)” was changed to “Specifically, the efficiency roll-off caused by phosphor-based triplet-triplet annihilation was significantly reduced in the two double-doped device, due to the singlet harvest through a long-range FRET pathway (Supplementary Fig. S14) and the subsequent fast radiative decay ($k_F = 1.2 \times 10^8 \text{ s}^{-1}$, Table 1).²¹ In addition, the small energy difference between the frontier orbitals of BCz-o-TRZ host (-5.26 eV/-2.64 eV, ref 38) and the tCzphBs (Table 1) avoids charge

carrier trapping and direct triplet exciton formation at the terminal dopant.”

6. Device lifetime is required to improve the quality of this work.

Author reply: See the reply to the last question of Reviewer #1.

Reviewer #3

In this study, the authors reported a novel strategy for achieving ultra-pure green emission by linking the outer phenyl groups in multiple-resonance (MR)-type B(boron)-N (nitrogen) molecules. It is impressive that the high EL efficiencies and high green color purity could be simultaneously obtained in the MR-OLEDs based on tCzphB-Ph and tCzphB-FI emitters. I think this manuscript can be accepted after minor revisions.

1. Why the two target molecules could emit ultra-pure green light? It's better to explain the design strategy in a prominent location.

Author reply: The FWHMs of the emission spectra are both 21 nm for tCzphB-Ph and tCzphB-FI in toluene, which is narrower than that of 25 nm for *t*-DABNA and DtBuCzB (Fig. 2b). The *t*-DABNA contains two isolated phenyl groups that are perpendicular to the BN-core plane, while the two phenyl groups are not isolated in tCzphB-Ph and tCzphB-FI but combined inside a skeleton with structure of indolo[3',2',1':8,1]quinolino[3,2-*b*]acridine. As a result, a higher rigid planar structure than *t*-DABNA and DtBuCzB is formed in tCzphB-Ph and tCzphB-FI. This contributes to suppression of not only the structural relaxation in the excited S₁ state which is responsible to the emission band but also the molecular vibrations.

Regarding the structural relaxation, small relaxation leads to small Huang-Rhys factor S value because of small shift of the equilibrium coordinate of the S₁ state potential from the S₀ state potential with the effective vibrational mode to form the 0-0, 0-1, 0-2 vibrational components in the whole PL spectrum. In fact the small S values close to 0.12 is obtained for tCzphB-Ph and tCzphB-FI. The influence of structural relaxation on spectral FWHM was illustrated in Supplementary Note 1.

To obtain narrowband emission, another factor is necessary in addition to the small S value, *i.e.* the 0-0 band itself should be narrow. The calculation of vibrational potential energy curves on the basis of the major vibrational mode indicated that the potential energy curve of the rigid tCzphBs is steeper than that of DtBuCzB (see the following figure). A narrow conformational distribution in the ground and excited states can narrow the 0-0 band of the emitter. We illustrated this concept in the calculation part of the text.

Supplementary Fig. 8. Vibrational potential energy curves of DtBuCzB and tCzphB-Ph on the basis of their major vibrational mode. Comparing the deformed geometric structures with the ground-state structure in Cartesian coordinates, the sum-modulus-difference for all atoms is recorded as a relative displacement.

The smallest FWHM of 14 nm is obtained in the PL spectrum measured in non-polar cyclohexane. When measured in other solvents with polar solvents, the FWHM is increased (Fig. 2c). The increase is caused mainly by the structural relaxation of the excited state of the solute by the dipole-dipole interaction with the solvent. The influence of solvation effect on spectral FWHM was illustrated in Supplementary Note 2.

Finally, in the Discussion Section of the manuscript, the following sentences were added to summarize the design strategies of a narrow-band emitter. “The combination of multiple resonance effect and rigid molecular framework suppresses both irreversible geometry relaxation and reversible stretching and rocking vibration in the excited state, which consequently minimized the Huang-Rhys factor (*i.e.* minimized the displacement of equilibrium position in the excited state) and narrowed the major vibrational band in the emission spectra.”

2. In Figure 3c, it seems that DtBuCzB possesses a smaller FWHM, please provide some comments on this point. In addition, to further explain the FWHM difference between these emitters, reorganization energy calculation should better be presented.

Author reply: Thank you for the suggestion. When we derive the vibration lines in Fig. 3 and Supplementary Fig. 7 by mean of Gaussian function with FWHM of 200 cm^{-1} , the FWHMs of the final spectra are 474 cm^{-1} for tCzphB-Ph, 440 cm^{-1} for tCzphB-F1 and 488 cm^{-1} for DtBuCzB. As the reviewer commented, the spectral FWHM of tCzphB-Ph is comparable to that of DtBuCzB. To explain the failure in the simulation, the following statement was added to the text. “The commonly used simulation method derives all vibration lines by Gaussian function with the same FWHM. However, the rigid tCzphBs exhibit a relatively steep vibrational potential energy curve compared to DtBuCzB (Supplementary Fig. 8), implying a narrower

conformational distribution in the ground and excited states. In Frank-Condon analyses of emission spectra, the FWHM used to resolve the major vibrational band for the tCzphBs should be narrower than that for DtBuCzB, which must further enlarge the difference between the FWHMs of the final spectra.”

We calculated the reorganization energies of the four investigated molecules and showed the result in Supplementary Fig. 6. Although the reorganization energies of the tCzphBs (0.13 and 0.12 eV) are lower than that of *t*-DABNA (0.17 eV), they are slightly higher than that of DtBuCzB (0.11 eV), indicating that the suppression of the high-frequency vibrations in tCzphBs is responsible for the narrowed 0-0 emission band.

Supplementary Fig. 6. Potential energy surfaces in the ground and excited states. The transition and reorganization energies were calculated by TD-DFT at the PBE0/6-31G (d, p) level in toluene.

3. The orbital calculation of the excited state is necessary to show the nature of the emission.

Author reply: The natural transition orbitals for the $S_1 \rightarrow S_0$ transition were calculated and shown in Fig. 3b and Supplementary Fig. 5.

4. The TADF characteristics such as Φ_F , Φ_{TADF} , k_{ISC} , and k_{RISC} of the novel emitters should better be calculated from the transient emissions.

Author reply: Thank you for the suggestion. The parameters of Φ_F , Φ_{TADF} , τ_F , τ_{TADF} and k_F have been calculated and listed in Table 1. Since k_{ISC} and k_{RISC} were not involved in the discussion, they were not listed in Table 1, but the readers can calculate them on their own using the parameters of Φ_F , Φ_{TADF} , τ_F and τ_{TADF} .

5. Why the emitters show larger FWHMs in OLEDs?

Author reply: For tCzphBs doped into the TPSS films, the FWHMs of the EL spectra

are only larger than those of the PL spectra by 1 nm, which can be attributed to the solvent effect of the blocking layer on the emitter. The employment of bipolar host BCz-o-TRZ and the introduction of the sensitizer Ir(ppy)₃ increases the polarity of the matrix surrounding the tCzphBs and therefore widen the EL spectra by the solid solvation effect.

6. How about of the stabilities of the OLEDs?

Author reply: See the reply to the last question of Reviewer #1.

REVIEWERS' COMMENTS

Reviewer #1 (Remarks to the Author):

This work was revised properly for publication.

Reviewer #2 (Remarks to the Author):

The authors have made significant and reasonable modifications according to the comments. This work could be published with its current version.

Reviewer #3 (Remarks to the Author):

All my concerns have been carefully addressed. The manuscript is acceptable as is.